# Mediterranean Diet and Risk of Dementia and Alzheimer’s Disease in the EPIC-Spain Dementia Cohort Study

**DOI:** 10.3390/nu13020700

**Published:** 2021-02-22

**Authors:** María Encarnación Andreu-Reinón, María Dolores Chirlaque, Diana Gavrila, Pilar Amiano, Javier Mar, Mikel Tainta, Eva Ardanaz, Rosa Larumbe, Sandra M. Colorado-Yohar, Fernando Navarro-Mateu, Carmen Navarro, José María Huerta

**Affiliations:** 1Section of Neurology, Department of Internal Medicine, Rafael Méndez Hospital, Murcia Health Service, 30813 Lorca, Murcia, Spain; 2Murcia Biomedical Research Institute (IMIB-Arrixaca), 30120 Murcia, Spain; mdolores.chirlaque@carm.es (M.D.C.); diana.gavrila@carm.es (D.G.); scyohar@gmail.com (S.M.C.-Y.); fernando.navarro@carm.es (F.N.-M.); carmen.navarro.100@gmail.com (C.N.); jmhuerta.carm@gmail.com (J.M.H.); 3Department of Epidemiology, Murcia Regional Health Council, 30008 Murcia, Spain; 4CIBER Epidemiología y Salud Pública (CIBERESP), 28029 Madrid, Spain; epicss-san@euskadi.eus (P.A.); me.ardanaz.aicua@navarra.es (E.A.); 5Department of Health and Social Sciences, University of Murcia, 30100 Murcia, Spain; 6Murcia Health Service, 30100 Murcia, Spain; 7Public Health Division of Gipuzkoa, Basque Government, 20013 Donostia-San Sebastián, Spain; 8Biodonostia Health Research Institute, 20014 San Sebastián, Spain; FRANCISCOJAVIER.MARMEDINA@osakidetza.eus; 9Clinical Management Unit, OSI Alto Deba, 20500 Arrasate-Mondragón, Spain; 10AP-OSIs Gipuzkoa Research Unit, OSI Alto Deba, 20500 Arrasate-Mondragón, Spain; 11Health Services Research Network on Chronic Patients (REDISSEC), 48902 Bilbao, Spain; 12CITA Alzheimer Foundation, 20019 Donostia-San Sebastián, Spain; mtainta@cita-alzheimer.org; 13Neurology Service, OSI Goierri-Alto Urola, 20700 Zumárraga, Spain; 14Public Health Institute of Navarra, IdiSNA, 31008 Pamplona, Spain; rosa.larumbe.ilundain@navarra.es; 15Neuroepigenetics Laboratory, Navarrabiomed, Public University of Navarre (UPNA), 31008 Pamplona, Spain; 16Department of Neurology, Complejo Hospitalario de Navarra, 31008 Pamplona, Spain; 17Research Group on Demography and Health, National Faculty of Public Health, University of Antioquia, Medellín 050010, Colombia; 18Unidad de Docencia, Investigación y Formación en Salud Mental (UDIF-SM), Servicio Murciano de Salud, 30120 Murcia, Spain

**Keywords:** Mediterranean diet, dementia, Alzheimer’s disease, cohort study, prospective analysis, EPIC-Spain

## Abstract

The Mediterranean diet (MD) has shown to reduce the occurrence of several chronic diseases. To evaluate its potential protective role on dementia incidence we studied 16,160 healthy participants from the European Prospective Investigation into Cancer and Nutrition (EPIC)-Spain Dementia Cohort study recruited between 1992–1996 and followed up for a mean (±SD) of 21.6 (±3.4) years. A total of 459 incident cases of dementia were ascertained through expert revision of medical records. Data on habitual diet was collected through a validated diet history method to assess adherence to the relative Mediterranean Diet (rMED) score. Hazard ratios (HR) of dementia by rMED levels (low, medium and high adherence levels: ≤6, 7–10 and ≥11 points, respectively) were estimated using multivariable Cox models, whereas time-dependent effects were evaluated using flexible parametric Royston-Parmar (RP) models. Results of the fully adjusted model showed that high versus low adherence to the categorical rMED score was associated with a 20% (HR = 0.80, 95%CI: 0.60–1.06) lower risk of dementia overall and HR of dementia was 8% (HR = 0.92, 0.85–0.99, *p* = 0.021) lower for each 2-point increment of the continuous rMED score. By sub-types, a favorable association was also found in women for non-AD (HR per 2-points = 0.74, 95%CI: 0.62–0.89), while not statistically significant in men for AD (HR per 2-points = 0.88, 0.76–1.01). The association was stronger in participants with lower education. In conclusion, in this large prospective cohort study MD was inversely associated with dementia incidence after accounting for major cardiovascular risk factors. The results differed by dementia sub-type, sex, and education but there was no significant evidence of effect modification.

## 1. Introduction

Dementia is one of the leading causes of disability and dependency among older people worldwide, with huge physical, social, and economic repercussions [1]. Over 50 million people suffer from dementia worldwide and this number is expected to rise by more than two-fold by 2050 [2], which has led the World Health Organization (WHO) to recognize dementia as a public health priority [3]. As an age-related disorder with no effective treatment to date [3], prevention through changes in modifiable risk factors and lifestyles such as diet remain paramount to tackle the health and social challenges imposed by the growing burden of dementia worldwide [4].

The Mediterranean diet (MD) pattern, described more than half a century ago, is characterized by a high intake of vegetables, fruits, legumes, unrefined cereals, nuts and olive oil, a moderate intake of fish and wine and low intake of dairy products, meat, poultry and saturated fat [5,6,7]. In the past few decades, the advantages of the MD have been extensively studied in relation to premature mortality and incidence of cancer or cardiovascular disease [8,9], while its health benefits have also led researchers to explore its potential role on cognition. Previous studies suggest that the MD may protect against age-related cognitive decline and mild cognitive impairment [10,11], but its potential role in reducing or delaying the onset of dementia is not well established [12,13,14,15,16]. Most studies conducted so far have focused on cognitive function as the end-point whereas only a few have assessed the potential beneficial role of the MD pattern on the occurrence of dementia or Alzheimer’s disease (AD), for which the level of the evidence is still regarded as moderate and warrants further research [2]. Moreover, many of the studies that have assessed the association between MD and dementia risk have taken place in non-Mediterranean countries, mostly the United States (US) [15,16,17]. These studies would support the external validity of the beneficial effect of the MD. However, there remains a need of supporting evidence from large-scale prospective studies carried out in Mediterranean countries in order to elucidate whether the MD could actually reduce the risk of cognitive decline or dementia. It is in these regions where the MD pattern remains closer to its original features. Moreover, the traditional lifestyles and healthy habits of Mediterranean countries could extend the benefits of the MD beyond its strictly nutritional effects. The EPIC (European Prospective Investigation into Cancer and nutrition)-Spain study is a large multicenter cohort study designed to evaluate the association between diet, lifestyles and incidence of cancer and other chronic and age-related diseases, such as dementia, accounting for wide exposure variability, long follow-up time and large number of cases. In this analysis we aimed to evaluate the relationship between adherence to a MD pattern and the subsequent risk of dementia and dementia sub-types in the EPIC-Spain Dementia Cohort.

## 2. Materials and Methods

### 2.1. Study Sample

The EPIC study is a multicenter prospective study carried out on over 500,000 volunteers from ten European countries [18,19]. The EPIC-Spain Dementia Cohort was established from a sample of 25,015 EPIC-Spain participants from three EPIC-Spain study centers which carried out the prospective ascertainment of dementia cases: Gipuzkoa, Navarra, and Murcia [20].

Participants were 30 to 70 years old at enrolment between 1992 and 1996, and were mostly blood donors, civil servants, and general population. Exclusion criteria were being pregnant or breastfeeding, or not being physically or mentally capable of participating. Baseline data on diet, lifestyles, and medical and reproductive history were collected during in-person interviews. In addition, participants underwent a physical examination to obtain anthropometric information, and provided a blood sample. Further details on study design and sample characteristics can be found elsewhere [19,20].

### 2.2. Dietary Assessment

Detailed information on habitual food consumption during the previous year was collected through a validated diet history method, administered face-to-face by trained dietitians [21].

Questionnaires were structured by meals according to occasions of food intake and subjects were asked about foods consumed in a typical week, accounting for food preparation, frequency of consumption and usual portion size. All foods consumed at least twice a month were considered (except liver, included when consumed at least once a month) and seasonal and weekly variations (working days or weekends) were taken into account. Total daily energy and nutrient intakes were calculated using country-specific food composition tables, further harmonized within the EPIC Nutrient DataBase project [22].

Plausibility of dietary intake was defined according to the predicted total energy expenditure (pTEE) method [23], identifying implausible reporters based on the ratio of estimated energy intake to predicted total energy expenditure (rEI:pTEE).

### 2.3. Relative Mediterranean Diet (rMED) Score

Adherence to the Mediterranean dietary pattern was assessed by means of the relative Mediterranean Diet score (rMED), a variation of the original Mediterranean diet score by Trichopoulou et al. [24] defined within the EPIC-Spain cohort.

The rMED score is based on 9 components: 6 considered as positive (fruit, vegetables, olive oil, legumes, fish, and cereals), 2 considered as negative or detrimental (meat and dairy products), and alcohol. Intake for each component was standardized as grams per 1000 kcal and divided by tertiles. Positive components score 0, 1, and 2 for the first, second, and third groups defined by tertiles of intake, respectively, while detrimental components score 2, 1, and 0 for the first, second, and third groups, respectively. For alcohol intake, 2 points are given to consumption within a sex-specific range (5–25 g/day for women, 10–50 g/day for men), and 0 points otherwise [24,25]. Thus, the rMED score ranged from 0 (minimum) to 18 (maximum). Levels of adherence to the MD were classified as low (0–6 points), medium (7–10 points), and high rMED scores (11–18 points) as previously defined [25].

Since associations of MD with health-related outcomes may vary depending on the operative definition of the MD pattern [6], three alternative MD indexes, i.e., the Mediterranean Diet Score (MDS) by Trichopoulou et al. [24], the adapted relative Mediterranean Diet score (arMED) by Buckland et al. [26], and the alternate Mediterranean Diet Index (aMED), by Fung et al. [27], were evaluated in a sensitivity analysis.

### 2.4. Ascertainment of Dementia Cases

A two-phase validation protocol was designed to ascertain incident dementia cases occurring in the cohort based on medical records, as detailed previously [28]. In brief, potential cases were identified by record linkage of the EPIC database with primary care records, hospital discharge databases, and regional mortality registries. Incident dementia cases were then validated after a careful and extensive examination of all medical records available for each potential case by a panel of neurologists, who determined the sub-type whenever possible.

Participants were followed up from recruitment until the date of diagnosis, death, loss to follow-up, or the last complete vital status check (31 December 2017 for Gipuzkoa, 31 December 2015 for Navarra, and 30 November 2016 for Murcia), whichever occurred first.

Participants in the EPIC study did not undergo a baseline cognitive assessment. However, all volunteers were considered to have normal cognition since they were required to be able to complete extensive and demanding questionnaires, including a dietary history interview which took 50–60 min on average. The case ascertainment process identified only one prevalent case, whereas four participants developed dementia within the first 5 years of the study (only one in the first 3 years).

From the 25,015 participants without evidence of prevalent dementia at baseline, we excluded a total of 1651 patients with at least one major chronic pathology (diabetes, ischemic heart disease, stroke, or cancer). We further excluded as energy mis-reporters a total of 7204 participants with a reported energy intake beyond 30% of the estimated energy requirement (corresponding to a cutoff of ±2 SD of the rEI:pTEE ratio), including 206 incident dementia cases. Thus, the final sample consisted of 16,160 participants, among which 308 incident AD cases and 151 incident non-AD (non-Alzheimer’s disease) cases were identified after a total observation time of 349,242 person-years, corresponding to a mean follow-up time (±SD) of 21.6 (±3.4) years).

### 2.5. Assessment of Anthropometric, Clinical and Lifestyle Data

Additional questionnaire information was collected at baseline on educational level, smoking status, and medical and reproductive history during in-person interviews. Data on prevalence of chronic diseases such as cancer, cardiovascular disease, or diabetes was self-reported. Height, weight, and waist and hip circumferences were measured by trained personnel during a physical examination following standard procedures [29]. Quetelet’s body mass index was estimated as weight (in kg) divided by squared height (in m).

### 2.6. Ethics

The EPIC study protocol was approved by the International Agency for Research on Cancer (IARC) Ethics Committee. All participants voluntarily agreed to take part and gave written informed consent. The current research has been conducted in accordance with the principles of the Declaration of Helsinki and the paper was written according to the Strengthening the Reporting of Observational Studies in Epidemiology (STROBE) statement (https://www.equator-network.org/reporting-guidelines/strobe) (accessed on 22 November 2020).

### 2.7. Statistics

Baseline characteristics of cases and non-cases were described using absolute and relative frequencies for categorical variables and medians and inter-quartile ranges for continuous variables. Statistical differences by case status or rMED categories were assessed with χ2 (categorical variables), Mann-Whitney U (for comparisons between two groups) or Kruskal-Wallis (for comparisons across three categories) tests.

Hazard ratios (HR) of dementia and dementia sub-types were estimated using proportional hazards Cox models, stratified by age (in 5-year categories) and study setting (to account for heterogeneity in methods and population characteristics across centers). Entry time was defined as age at recruitment, and exit time was age at diagnosis for dementia cases, and age at death or censoring for non-cases. Adherence levels were defined based on the rMED, by categorizing the score into low (0–6 points), medium (7–10 points) and high (11–18 points) adherence groups. HR were estimated for ‘medium’ and ‘high’ rMED categories as compared with ‘low’ rMED scores, and for every 2-point increment of the score as a continuous variable under the linearity assumption. Besides, non-linear associations of dementia with the rMED score were evaluated after a restricted cubic spline transformation of the exposure (with internal knots at the 33rd and 67th centiles), by plotting the predicted HR against the rMED score. A basic model was fitted including sex, educational level, and energy intake as potential confounders, whereas final multivariable models were further adjusted by other variables which could partially account for confounding of the exposure–outcome association: smoking habit (never, former, current, unknown), BMI group (normal weight, overweight, obese), elevated waist circumference (≥102 cm in men, ≥88 cm in women), sum of household and recreational physical activity (in Metabolic equivalents [MET]∙h/week), self-reported hypertension (no, yes, unknown), self-reported hyperlipidemia (no, yes, unknown), combined coffee and tea consumption (in mL/day), and daily intake of potatoes, eggs, and cakes and biscuits, in (g/day). In women, final multivariable models also included menopausal status (pre-, peri-, post-menopausal), ever use of oral contraceptives (yes, no, unknown), and ever use of hormonal replacement therapy (yes, no, unknown). The proportional hazards assumption was checked in all models based on Schoenfeld residuals and visual inspection of log-log survival plots, and no significant deviations were found.

Main analyses were stratified by sex and dementia sub-types (AD and non-AD), whereas potential effect modification of the rMED and dementia association was evaluated for sex, educational level, smoking habit, and obesity. Likelihood ratio tests were used to assess heterogeneity.

Time-dependent effects were evaluated using flexible parametric Royston-Parmar (RP) multivariable models [30], by plotting the HR for high vs. low rMED scores over follow-up time. RP models implement the use of separate sets of spline terms to model baseline hazard rates and time-dependent effects of a covariate, thus allowing for a smooth representation of the change in HR over time.

All analyses were performed with Stata/SE v.14.2 (StataCorp LLC, College Station, TX, USA). Two-sided (when appropriate) *p*-values < 0.05 were considered statistically significant.

## 3. Results

459 incident cases of dementia (67% AD) were available for analysis after a mean of 21.6 (±3.4) years (see Appendix A, flowchart of study participants). Baseline characteristics of the sample, including dementia and AD cases, are summarized in Table 1.

Dementia cases were more likely to be older, obese, to have low educational level, higher intake of fruits and dairy products and lower intake of meat. On the contrary, non-cases were more likely to be smokers, and to consume more energy, alcohol, coffee and tea. High adherence to the rMED score was positively associated to age, male sex, obesity and intake of total energy and nuts (Appendix A). High adherence was also more frequent among post-menopausal women. By contrast, low adherence to the rMED was associated with smoking, leisure time physical activity, and higher intake of eggs, coffee and tea.

Table 2 shows the results of the multivariable survival analyses between rMED adherence and incidence of dementia, overall and by sex. A greater adherence to the MD pattern was associated with a 20% lower risk of dementia (HR = 0.80, 95%CI: 0.60–1.06), and an estimated 8% lower risk per 2-point increment in the continuous rMED score (*p* for linear trend on the continuous variable = 0.021). The negative trend was found to be statistically significant in women, but not men. The shape of the association is graphically illustrated in Figure 1, which shows the restricted cubic spline modelling of dementia risk by sex according to the rMED score, suggesting a non-linear relationship with a steeper slope at lower rMED scores. Although point estimates were similar in both sexes, results in men did not reach statistical significance.

When further stratifying results by dementia sub-types, the negative association with MD was stronger among women for non-AD dementia, with up to 48% lower risk (95%CI: 0–73%) for those with high vs. low adherence to the MD, whereas rMED suggest a favorable while not statistically significant association with AD (HR high vs. low rMED = 0.57, 95%CI: 0.32–1.01) among men (Table 3). The decline in dementia risk was steeper for non-AD in the low-medium score range of the rMED, whereas the favorable association with AD risk suggested no evidence against a linear relationship (*p* for non-linearity = 0.353, Appendix A).

Table 4 evaluates selected variables as potential effect modifiers of the rMED and dementia association. The results, while not statistically significant, suggest a possible heterogeneity for education (*p* for interaction = 0.055). Among those with primary education or lower, participants with medium and high adherence to the MD had about 20% lower risk of dementia, a pattern that was not consistent in participants with higher education. The rMED score was significantly associated with dementia in women and non-smokers, but there was no formal evidence of heterogeneity by sex, smoking or ponderal status.

Time-varying effects for the association of rMED with dementia were studied by means of RP flexible survival models, by plotting HR as a function of follow-up time (Figure 2). Results show the change in HR estimates throughout study time, revealing that HR < 1 were detectable only after a long follow-up (~18 years), when the larger cumulative number of cases, which increased exponentially with the aging of the sample, allowed for more powerful estimations.

The distribution of baseline characteristics and dietary variables according to rMED categories are shown in Table 5, separately for cases and non-cases. The rMED score was effective in ranking participants according to their dietary intake for almost all the diet groups included in the index (except for nuts and seeds, eggs, and coffee). Besides, higher adherence to the rMED score was associated with older age, female sex, lower prevalence of smoking, higher prevalence of obesity and less leisure time physical activity among non-cases. For cases, differences were only observed for sex and leisure time physical activity. Notably, there were no differences in educational level by rMED categories in any group.

Further supplementary analyses showed the robustness of the results to exclusion of components of the rMED score on an item-by-item basis (Appendix A), to different multivariable models with increasing levels of adjustment, to other sensitivity analyses to account for potential reverse causation (Appendix A), and to alternative operational definitions of the MD (arMED, MDS, and aMED scores) (Appendix A). Stratification of the main analysis according to plausibility of energy reporting (Appendix A), which would eventually lead to misclassification of dietary exposure, resulted in wide confidence intervals and non-significant dementia risk estimates among mis-reporters (*p* for linear trend = 0.887).

## 4. Discussion

In this large prospective study involving 16,160 Spanish middle-aged and elderly participants followed up for over 20 years, participants with ‘high’ adherence compared to those with ‘low’ adherence to the MD (categorical rMED score) had a 20% lower risk of dementia overall. A negative linear trend was significant among women, those with lower educational level, and non-smokers. By dementia sub-types, associations were stronger for non-AD dementia in women and for AD in men. Of note, the association between rMED score and dementia incidence was revealed only after mis-reporters of energy intake were excluded from the analyses.

Mis-reporting (over- or under-reporting) of energy intake is a potential source of error in nutritional epidemiological studies, affecting the reliability and validity of nutritional assessment and confounding or attenuating diet–disease associations [31]. Different methods have been developed to minimize the impact of mis-reporting bias in epidemiological studies that rely on the ratio of reported intakes to predicted total energy expenditure, estimating a reference interval to account for individual variations in physical activity levels (PAL) [32,33]. Reporting of dietary energy can thus be regarded as plausible when the energy intake calculated from the dietary questionnaire falls within the defined cut-offs, or implausible (over- or under-reporting) otherwise. All EPIC-Spain participants have been assessed for reporting plausibility following the methodology described by Mendez et al. [23], and classified accordingly. Our data suggested that dietary mis-reporting had a non-negligible impact in the accuracy of estimates of dementia risk, increasing estimation errors and reducing the power to detect significant associations. Therefore, we have restricted the main analyses to plausible reporters.

There is suggestive evidence for a protective role of the MD with regards to dementia risk [2], and a considerable body of evidence suggests that MD may have a protective effect on cognition by decreasing the risk of cognitive impairment and delaying the onset of dementia [10,11,34]. In agreement with previous findings, our results suggest that adherence to MD could significantly decrease the risk of dementia [10,11]. Every 2-point increase of the rMED score was associated with an 8% and 6% lower risk of dementia and AD (although results for AD were not statistically significant), respectively, and results suggest a dose-response effect with decreasing risks of dementia at higher scores of the rMED score. However, a protective effect of the Mediterranean dietary pattern against cognitive decline or dementia has been reported in some [13,17,35,36,37,38] but not all [12,35] previous studies. Several cohort studies showed that the MD or its components were positively associated with a better cognition and contributed to delay the onset of mild cognitive impairment (MCI), both in Mediterranean [13,39,40] and non-Mediterranean countries [41,42]. Results from the PREDIMED (PREvención con DIeta MEDiterránea) clinical trial also support the role of MD and olive oil consumption on cognitive performance and the decreased risk of age-related cognitive decline [38]. However, only a few prospective cohort studies have examined the association between MD and the incidence of dementia or AD [12,13,17,37], and the available evidence in support of a protective role of MD remains inconclusive. In the WHICAP (Washington Heights-Inwood Columbia Aging Project) study, Scarmeas et al. [17] found that AD risk decreased by 9% for each additional point of the MeDi score (HR = 0.91; 95%CI: 0.83, 0.98), a MD scale ranging from 0 to 9 points, among 2258 US non-demented elderly followed up for a mean of 4 years. Furthermore, participants in the upper (vs. lower) scoring third had a 40% lower risk for AD (HR = 0.60, 95%CI: 0.42, 0.87). A further study by Morris et al. [16] in 923 US elderly participants from the Memory and Aging Project (MAP) project observed a 54% lower risk of AD when comparing participants in the high and low groups based on tertiles of the MIND (Mediterranean-Dietary Approach to Systolic Hypertension (DASH) diet intervention for neurodegenerative delay (MIND) diet) score (HR = 0.46; 95%CI: 0.27, 0.79), a hybrid pattern that combined the MD and the DASH diet. Our result of a negative trend in dementia risk for the continuous rMED score thus add to previous data in support of a beneficial role of MD against dementia. However, a recent large US prospective study conducted by Hu et al. in 13,630 participants from the Atherosclerosis Risk in Communities (ARIC) Study found no significant associations for the aMED, Healthy Eating Index (HEI)-2010 or DASH dietary scores with incident dementia after 27 years of follow-up [37]. This is in line with a previous French study by Feart et al. [12], who followed a cohort of 1410 French elderly over 5 years and could not find evidence that MD decreased dementia risk, despite a significant association with lower cognitive decline as defined by higher MMSE (Mini-Mental State Examination) scores. Of note, the authors acknowledged the limited power to detect significant effects given the scarce number of cases and short follow-up time. Nevertheless, in spite of the scarce prospective literature available on the association of MD with dementia or AD risk, our results add to previous evidence in support of a beneficial role for the MD against dementia [10,11,43,44].

The MD could be related to a lower risk of cognitive decline and dementia through several potential mechanisms, including anti-inflammatory, antioxidant and lipid-lowering actions [45] and a favorable effect on cardiovascular risk factors [44,46]. The MD has been associated with lower levels of C-reactive protein and interleukins, previously found to increase the risk of major forms of dementia such as AD and vascular dementia [47,48]. The MD has also been shown to modulate glucose and lipid metabolism, improving insulin sensitivity and blood lipid profile, and to reduce the risk for hypertension, diabetes, or cardiovascular disease [8,9,45], risk factors for cognitive decline and dementia [4]. Furthermore, several components of the MD, such as olive oil, fish, vegetables, or nuts are associated with reduced inflammation and dyslipidemia [46,49], and are rich in antioxidants, such as vitamin C, vitamin E, β-carotene, or flavonoids, and minerals such as selenium [45], that could ameliorate the age-related decline in cognitive function that precedes the onset of dementia. The pleiotropy of the advantageous effects of the MD supports the biological plausibility of its potential role in reduction of dementia risk.

We found no evidence that the association of rMED with dementia incidence varied by sex, smoking or obesity, which are potential effect modifiers as risk factors for dementia or cardio-metabolic risk that configure the patho-physiological environment (i.e., inflammation, oxidative stress, neurovascular dysfunction) through which diet would exert its effects on the prevention of dementia or dementia sub-types [44,45,46]. However, there was some indication of heterogeneity according to educational level (P interaction = 0.055), so that estimates were more robust among participants with lower education across categories of the rMED score. However, considering the similarity in HR estimates for the high vs. low rMED categories, as well as for the score as a continuous variable, the heterogeneity reported might be a consequence of the scarce number of events that occurred in the group with higher educational level, distorting the estimates in the medium score group. Our results were robust against a series of sensitivity analyses. The case-wise exclusion of single rMED components had little influence on overall estimates (Appendix A). We further excluded the first five years of follow-up and found no evidence of a potential reverse causation bias (Appendix A). We also considered alternative operational definitions of the MD, as the use of different scores might partly account for the heterogeneous results in the literature. We found a similar association in shape and magnitude for the rMED and arMED indices, which was slightly attenuated when using the MDS, whereas the aMED score showed a flat null association with dementia risk in our cohort. Of note, the aMED index was developed by Fung et al. as an attempt to adapt the MDS to the US population [27].

Our study has some limitations. The EPIC cohort was established to study the determinants of cancer and it was not designed to study dementia as a primary outcome. For this reason, participants did not undergo a baseline cognitive assessment to exclude prevalent cases of dementia or mild cognitive impairment. However, only one case was validated as prevalent after the revision of clinical records of potential cases, and the exclusion of participants with less than five years of follow-up had no impact on the estimates. Furthermore, diagnoses were based on clinical records available from the public health system, rather than prospective evaluations of the participants’ cognitive function. Nevertheless, given the universal coverage of the country health system, we are confident that virtually all incident cases from our cohort diagnosed with dementia have been ascertained. Other limitations are that diet and covariate data were only assessed at baseline and we could not evaluate potential exposure and lifestyle changes during follow-up, and that we had no data available on genetic traits such as apolipoprotein E (APOE) genotype (presence of ε4 alleles), although previous evidence suggest that APOE is not associated with MD and has no influence on the association between MD and AD [17]. As in any observational study, we cannot rule out the possibility of residual or unmeasured confounding. Finally, the lack of representativeness of the cohort could limit the generalizability of the results to other populations.

Major strengths of our study are the large sample size and long follow-up that were revealed as crucial for the ability to identify potential clinically relevant associations with sufficient precision. Nevertheless, statistical power could still be limited in some sub-group analyses with lower number of cases, as reflected by wider confidence intervals. Of note, this is the second largest study available evaluating the association of MD with a hard dementia or AD endpoint, and the largest ever conducted in a Mediterranean country. Lack of sufficient number of cases or an insufficient follow-up time could have prevented other studies from reaching significant conclusions. The Mediterranean setting of the cohort, its geographical variability within the country, and the availability of an extensive set of socio-demographic, lifestyle, and clinical variables to adjust for are important features. Finally, mis-reporting was shown to be an important factor to account for. Therefore, being able to control mis-reporters by identifying and excluding them from the analysis reinforced the accuracy of our estimates.

## 5. Conclusions

In conclusion, adherence to the Mediterranean diet was associated with a 20% lower risk of dementia overall in the EPIC-Spain Dementia Cohort, a Mediterranean study involving over 16,000 middle-aged and elderly participants followed up for over 20 years. Associations were stronger for non-AD dementia in women and for AD in men, and among participants with lower education. The significant associations were revealed after excluding energy mis-reporters and required a long follow-up time. Further cohort studies with sufficient number of cases and follow-up will contribute to reinforce the evidence for the role of MD in risk reduction of cognitive decline, dementia, and Alzheimer’s disease.

## Figures and Tables

**Figure 1 nutrients-13-00700-f001:**
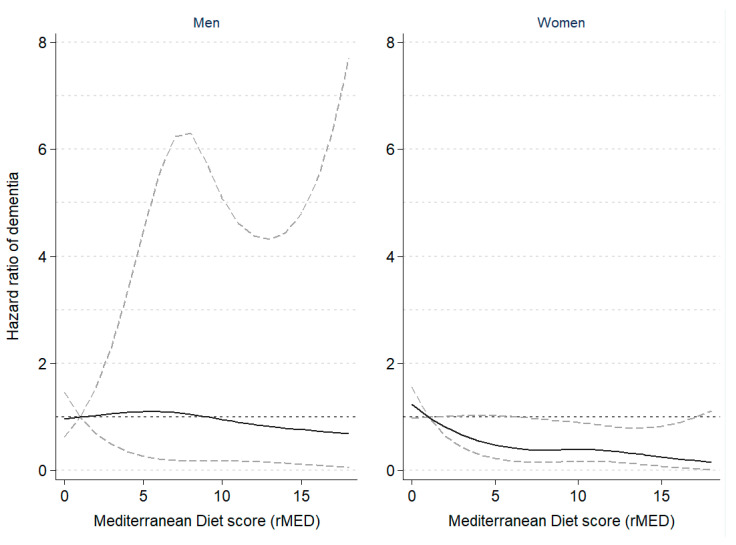
Hazard ratio of dementia according to Mediterranean Diet scores in the EPIC-Spain Dementia Cohort study (*N* = 16,160), by sex. The dashed lines represent the upper and lower 95% confidence interval limits of the estimates. A non-linear inverse association between risk of dementia and rMED scores was observed among women, but not among men. Hazard ratios of dementia were estimated using Cox proportional hazards regression models, with age as the time scale, stratified by center and age (in 5-year categories), and adjusted by sex, education, energy intake, smoking, BMI category, elevated waist circumference, household and recreational physical activities, hypertension (self-reported), hyperlipidemia (self-reported), coffee and tea consumption (combined), and intake (in g/day per 2000 kcal) of potatoes, eggs, and cakes and biscuits. In women, models were further adjusted by menopausal status, oral contraceptive use, and hormone replacement therapy. Dementia risk was modelled following a restricted cubic spline transformation of the rMED variable with three degrees of freedom (knots were placed at the 33rd and 67th percentiles).

**Figure 2 nutrients-13-00700-f002:**
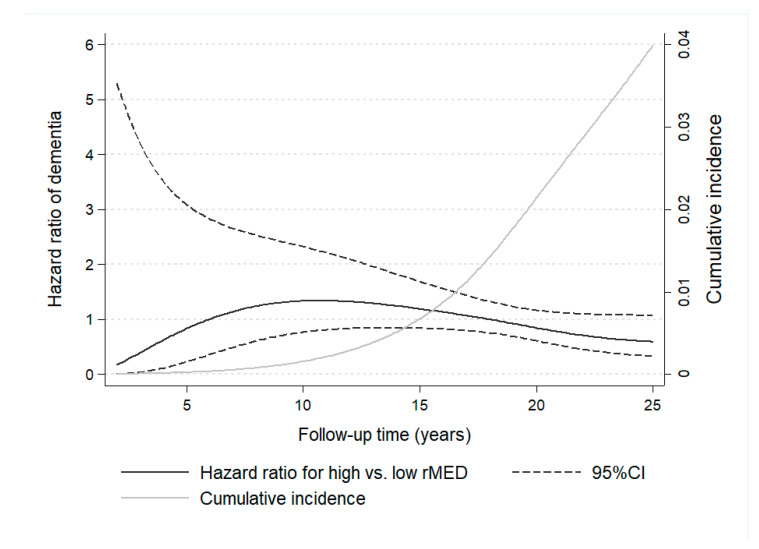
Time-dependent variation in hazard ratio estimates of dementia for participants with high versus low Mediterranean Diet scores (rMED) throughout 25 years of follow-up in the EPIC-Spain Dementia Cohort study (*N* = 16,160). Hazard ratio estimates of dementia varied depending on follow-up time and cumulative number of cases. Time-varying hazard ratios were calculated using flexible parametric Royston-Parmar survival models, with time on study as the time scale. Models were adjusted by center, sex, education, energy intake, smoking, BMI category, elevated waist circumference, household and recreational physical activities, hypertension (self-reported), hyperlipidemia (self-reported), coffee and tea consumption (combined), and intake (in g/day per 2000 kcal) of potatoes, eggs, and cakes and biscuits.

**Table 1 nutrients-13-00700-t001:** Baseline characteristics of cases (overall dementia and Alzheimer’s disease, AD) and non-cases participating in the European Prospective Investigation into Cancer and Nutrition (EPIC)-Spain Dementia Cohort study (*N* = 16,160).

	Non-Cases (*n* = 15,701)	Dementia Cases (*n* = 459)	AD Cases (*n* = 308)	*p* for Cases vs. Non-Cases	*p* for AD vs. Non-Cases
Age (y)	48.3 (12.1)	59.6 (7.5)	59.9 (7.2)	<0.001	<0.001
Female sex	8,974 (57.2%)	265 (57.7%)	187 (60.7%)	0.805	0.211
Low educational level ^1^	11,215 (71.4%)	407 (88.7%)	273 (88.6%)	<0.001	<0.001
Smoking				<0.001	<0.001
Never	8253 (52.6%)	306 (66.7%)	216 (70.1%)		
Former	2884 (18.4%)	68 (14.8%)	38 (12.3%)		
Current	4555 (29.0%)	85 (18.5%)	54 (17.5%)		
Overweight or obese	12,139 (77.3%)	402 (87.6%)	271 (88.0%)	<0.001	<0.001
Elevated waist circumference ^2^	6509 (41.5%)	278 (60.6%)	186 (60.4%)	<0.001	<0.001
Leisure time physical activity (MET ^3^·h/week) ^4^	86.6 (92.5)	91.8 (93.0)	96.1 (93.8)	0.105	0.0635
Energy intake (kcal/day)	2062.4 (673.3)	1964.2 (685.8)	1931.9 (678.2)	<0.001	<0.001
Potatoes (g/day per 2000 kcal)	72.1 (54.7)	72.1 (56.6)	68.9 (59.5)	0.598	0.959
Vegetables (g/day per 2000 kcal)	235.9 (180.4)	236.1 (174.6)	240.1 (178.8)	0.533	0.548
Fruits (g/day per 2000 kcal)	278.6 (270.9)	321.2 (278.4)	336.9 (300.7)	<0.001	<0.001
Legumes (g/day per 2000 kcal)	42.8 (38.1)	44.1 (40.6)	41.6 (39.9)	0.363	0.598
Fish and seafood (g/day per 2000 kcal)	52.0 (44.9)	55.4 (47.3)	57.8 (48.8)	0.057	0.022
Cereals (g/day per 2000 kcal)	180.1 (93.6)	178.3 (98.3)	177.3 (99.2)	0.947	0.565
Olive oil (g/day per 2000 kcal)	20.4 (20.6)	20.0 (21.1)	20.5 (22.1)	0.304	0.990
Nuts and seeds (g/day per 2000 kcal)	0.0 (3.5)	0.0 (2.9)	0.0 (2.2)	0.769	0.032
Meat (g/day per 2000 kcal)	120.2 (60.5)	117.2 (60.1)	116.9 (59.4)	0.025	0.044
Dairy products (g/day per 2000 kcal)	247.6 (217.8)	273.5 (249.6)	278.8 (257.2)	0.007	0.0021
Eggs (g/day per 2000 kcal)	22.6 (22.2)	21.2 (25.2)	21.0 (25.8)	0.181	0.290
Coffee and tea (mL/day)	100.0 (157.1)	62.9 (146.0)	63.9 (146.0)	<0.001	<0.001
Alcohol (g/day)	4.6 (20.2)	3.2 (19.1)	2.0 (14.5)	0.016	0.002
Mediterranean diet score (rMED)	9.0 (4.0)	9.0 (4.0)	9.0 (4.0)	0.539	0.617

Numbers are counts and percentages, or medians and inter-quartile ranges (IQR). *p* values obtained with Mann-Whitney U tests for continuous variables or X^2^ tests for categorical ones. ^1^ Primary studies or less. ^2^ Waist circumference ≥102 cm (men) or ≥88 cm (women). ^3^ MET: Metabolic equivalents. ^4^ Sum of recreational and household physical activities.

**Table 2 nutrients-13-00700-t002:** Hazard ratio of dementia by levels of the Mediterranean Diet score (rMED) in participants from the EPIC-Spain Dementia Cohort study (*N* = 16,160).

			Model 1			Model 2		
Adherence to the rMED	Person-Years	Cases	HR (95% CI)			HR (95% CI)		
**All**								
rMED categorical Low	68,876	85		1			1	
Medium	181,469	235	0.85 (0.66, 1.09)			0.85 (0.66, 1.09)		
High	98,957	139	0.79 (0.59, 1.04)			0.80 (0.60, 1.06)		
rMED Continuous (per 2-point increment)			0.91 (0.85, 0.98)			0.92 (0.85, 0.99)		
*p* for linear trend *			0.012			0.021		
*p* for non-linear trend *			0.063			0.094		
**Women**								
rMED categorical Low	47,718	59		1			1	
Medium	106,420	137	0.85 (0.63, 1.16)			0.84 (0.61, 1.15)		
High	47,884	69	0.88 (0.61, 1.25)			0.87 (0.60, 1.26)		
rMED Continuous (per 2-point increment)			0.91 (0.83, 1.00)			0.90 (0.82, 1.00)		
*p* for linear trend *			0.042			0.040		
*p* for non-linear trend *			0.077			0.086		
**Men**								
rMED categorical Low	21,158	26		1			1	
Medium	75,049	98	0.80 (0.52, 1.24)			0.81 (0.52, 1.26)		
High	51,073	70	0.65 (0.41, 1.03)			0.69 (0.43, 1.09)		
rMED Continuous (per 2-point increment)			0.91 (0.82, 1.02)			0.93 (0.83, 1.03)		
*p* for linear trend *			0.092			0.174		
*p* for non-linear trend *			0.400			0.579		

* *p*-values for linear and non-linear trend on the continuous variable. HR, hazard ratio; CI, confidence interval. *p*-values <0.05 or 95% CI that did not include the null value (i.e., 1) were considered statistically significant. Model 1: Cox regression adjusted by sex, education, and energy intake, and stratified by center and age. Model 2: As model 1, plus smoking, BMI category, elevated waist circumference, household and recreational physical activities, hypertension (self-reported), hyperlipidemia (self-reported), coffee and tea consumption (combined), and intake (in g/day per 2000 kcal) of potatoes, eggs, and cakes and biscuits. Women-specific model further adjusted by menopausal status, use of oral contraceptives (ever) and hormone replacement therapy (ever). MD adherence levels defined as low: 0–6 points, medium: 7–10 points, and high: 11–18 rMED score points.

**Table 3 nutrients-13-00700-t003:** Hazard ratio of dementia sub-types (Alzheimer’s and non-Alzheimer’s disease) by levels of Mediterranean Diet score (rMED) in participants from the EPIC-Spain Dementia Cohort study (*N* = 16,160).

		AD				Non-AD			
Adherence to the rMED	Person-Years	Cases	HR (95% CI)			Cases	HR (95% CI)		
**All**									
rMED categorical Low	68,876	55		1		30		1	
Medium	181,469	165	0.98 (0.71, 1.33)			70	0.64 (0.41, 0.99)		
High	98,957	88	0.87 (0.61, 1.24)			51	0.68 (0.42, 1.10)		
rMED Continuous (per 2-point increment)			0.94 (0.86, 1.03)				0.87 (0.77, 0.99)		
*p* for linear trend *			0.196				0.037		
*p* for non-linear trend *			0.353				0.097		
**Women**									
rMED categorical Low	47,718	37		1		22		1	
Medium	106,420	101	1.04 (0.71, 1.53)			36	0.52 (0.30, 0.89)		
High	47,884	49	1.11 (0.70, 1.74)			20	0.52 (0.27, 1.00)		
rMED Continuous (per 2-point increment)			0.98 (0.88, 1.10)				0.74 (0.62, 0.89)		
*p* for linear trend *			0.775				0.001		
*p* for non-linear trend *			0.660				0.003		
**Men**									
rMED categorical Low	21,158	18		1		8		1	
Medium	75,049	64	0.78 (0.46, 1.32)			34	0.89 (0.41, 1.95)		
High	51,073	39	0.57 (0.32, 1.01)			31	0.95 (0.43, 2.11)		
rMED Continuous (per 2-point increment)			0.88 (0.76, 1.01)				1.01 (0.85, 1.21)		
*p* for linear trend *			0.071				0.884		
*p* for non-linear trend *			0.343				0.816		

* *p*-values for linear and non-linear trend on the continuous variable. AD, Alzheimer’s disease; HR, hazard ratio; CI, confidence interval. *p*-values <0.05 or 95% CI that did not include the null value (i.e., 1) were considered statistically significant. Hazard ratios estimated using center- and age- stratified Cox regression models, adjusted by sex, education, energy intake, smoking, BMI category, elevated waist circumference, household and recreational physical activities, hypertension (self-reported), hyperlipidemia (self-reported), coffee and tea consumption (combined), and intake (in g/day per 2000 kcal) of potatoes, eggs, and cakes and biscuits. Women-specific model further adjusted by menopausal status, use of oral contraceptives (ever) and hormone replacement therapy (ever). MD adherence levels defined as low: 0–6 points, medium: 7–10 points, and high 11–18 rMED score points.

**Table 4 nutrients-13-00700-t004:** Hazard ratios of dementia by levels of adherence to the Mediterranean Diet score (rMED), stratified by selected variables in participants from the EPIC-Spain Dementia Cohort study (*N* = 16,160).

			rMED Score					
			Low	Medium	High	Continuous (per 2 Units)	*p* for Trend	*p* for Interaction
Group	Person-Years	Cases	(Ref.)	HR (95% CI)	HR (95% CI)			
All	349,302	459	1	0.85 (0.66, 1.09)	0.80 (0.60, 1.06)	0.92 (0.85, 0.99)	0.021	
Women	202,022	265	1	0.84 (0.61, 1.15)	0.87 (0.60, 1.26)	0.90 (0.82, 0.99)	0.038	0.527
Men	147,280	194	1	0.81 (0.52, 1.26)	0.69 (0.43, 1.09)	0.93 (0.83, 1.03)	0.174	
Primary education	249,720	407	1	0.80 (0.61, 1.05)	0.81 (0.60, 1.10)	0.92 (0.85, 1.00)	0.039	0.055
Secondary or higher	99,581	52	1	1.57 (0.67, 3.68)	0.88 (0.34, 2.31)	0.92 (0.74, 1.14)	0.448	
Non-smokers ^1^	250,380	374	1	0.88 (0.66, 1.16)	0.81 (0.59, 1.12)	0.92 (0.84, 0.99)	0.034	0.783
Smokers	98,922	85	1	0.71 (0.38, 1.32)	0.77 (0.40, 1.49)	0.94 (0.80, 1.10)	0.440	
Non-obese ^2^	196,071	164	1	0.92 (0.60, 1.40)	0.67 (0.41, 1.09)	0.90 (0.80, 1.01)	0.079	0.306
Obese	153,231	295	1	0.80 (0.58, 1.10)	0.85 (0.59, 1.21)	0.92 (0.84, 1.01)	0.083	

^1^ Never or former smokers. ^2^ Body mass index <30 kg/m^2^ and waist circumference <102 cm (men) or <88 cm (women). Hazard ratios estimated using center- and age- stratified Cox regression models, adjusted by sex, education, energy intake, smoking, BMI category, elevated waist circumference, household and recreational physical activities, hypertension (self-reported), hyperlipidemia (self-reported), coffee and tea consumption (combined), and intake (in g/day per 2000 kcal) of potatoes, eggs, and cakes and biscuits. MD adherence levels defined as low: 0–6 points, medium: 7–10 points, and high 11–18 rMED score points.

**Table 5 nutrients-13-00700-t005:** Baseline characteristics of dementia cases and non-cases by rMED score group in the EPIC-Spain Dementia Cohort study (N = 16,160).

	Non-Cases (*n* = 15,701)							Cases (*n* = 459)						
	Low (*n* = 3114)		Medium (*n* = 8163)		High (*n* = 4424)		*p*	Low (*n* = 85)		Medium (*n* = 235)		High (*n* = 139)		*p*
Age (y)	47.1	(11.6)	48.1	(12.0)	49.6	(12.0)	<0.001	59.4	(8.6)	60.0	(7.6)	59.1	(6.9)	0.431
Female sex	2137	(68.6%)	4725	(57.9%)	2112	(47.7%)	<0.001	59	(69.4%)	137	(58.3%)	69	(49.6%)	0.014
Low educational level ^1^	2265	(72.7%)	5832	(71.4%)	3118	(70.5%)	0.102	78	(91.8%)	203	(86.4%)	126	(90.7%)	0.276
Smoking							<0.001							0.574
Never	1644	(54.1%)	4337	(53.2%)	2232	(50.5%)		58	(68.2%)	163	(69.4%)	85	(61.2%)	
Former	429	13.8%)	1483	(18.2%)	972	(22.0%)		12	(14.1%)	33	(14.0%)	23	(16.6%)	
Current	999	(32.1%)	2339	(28.7%)	1217	(27.5%)		15	(17.7%)	39	(16.6%)	31	(22.3%)	
Overweight or obese	2327	(74.7%)	6309	(77.3%)	3503	(79.2%)	<0.001	77	(90.6%)	198	(84.3%)	127	(91.4%)	0.085
Elevated waist circumference ^2^	1288	(41.4%)	3423	(41.9%)	1798	(40.6%)	0.371	48	(56.5%)	143	(60.9%)	87	(62.6%)	0.656
Leisure time physical activity (MET ^3^·h/week) ^4^	99.8	(95.0)	86.6	(92.9)	78.1	(87.2)	<0.001	117.3	(119.5)	91.5	(88.2)	85.3	(82.5)	0.013
Energy intake (kcal/day)	2007.5	(625.0)	2055.3	(684.0)	2120.4	(673.8)	<0.001	1,909.5	(487.0)	1,964.2	(692.4)	2,029.8	(695.9)	0.724
Potatoes (g/day per 2000 kcal)	63.9	(52.7)	72.2	(54.0)	77.4	(55.3)	<0.001	61.2	(41.8)	73.4	(52.1)	83.3	(59.9)	0.002
Vegetables (g/day per 2000 kcal)	159.6	(124.6)	232.0	(167.4)	306.7	(179.9)	<0.001	156.2	(108.2)	236.1	(154.0)	295.3	(177.7)	<0.001
Fruits (g/day per 2000 kcal)	188.9	(224.5)	276.5	(262.8)	356.2	(256.5)	<0.001	212.8	(248.6)	327.0	(269.5)	366.3	(278.5)	<0.001
Legumes (g/day per 2000 kcal)	28.6	(29.1)	41.9	(35.1)	56.5	(39.0)	<0.001	28.3	(33.8)	41.4	(35.1)	62.1	(46.8)	<0.001
Fish and seafood (g/day per 2000 kcal)	34.9	(30.2)	550.5	(41.5)	72.0	(48.4)	<0.001	38.2	(40.0)	54.6	(43.5)	68.4	(53.0)	<0.001
Cereals (g/day per 2000 kcal)	146.9	(82.9)	178.5	(92.5)	206.0	(82.0)	<0.001	142.4	(87.9)	175.8	(89.9)	216.7	(103.3)	<0.001
Olive oil (g/day per 2000 kcal)	10.8	(20.0)	19.6	(20.2)	27.0	(14.7)	<0.001	9.9	(16.6)	18.4	(22.2)	26.2	(10.6)	<0.001
Nuts and seeds (g/day per 2000 kcal)	0.0	(2.7)	0.0	(3.2)	0.3	(4.2)	<0.001	0.0	(2.2)	0.0	(2.9)	0.3	(3.8)	0.235
Meat (g/day per 2000 kcal)	140.9	(57.6)	123.5	(59.1)	101.1	(52.4)	<0.001	137.5	(50.6)	121.4	(59.8)	96.7	(48.0)	<0.001
Dairy products (g/day per 2000 kcal)	359.4	(246.8)	254.4	(210.5)	183.3	(162.9)	<0.001	385.3	(214.2)	276.6	(249.6)	184.0	(210.9)	<0.001
Eggs (g/day per 2000 kcal)	23.7	(23.5)	23.2	(22.6)	20.7	(20.7)	<0.001	19.4	(20.8)	21.8	(27.0)	21.5	(25.1)	0.446
Coffee and tea (mL/day)	109.8	(166.8)	100.0	(157.5)	86.7	(143.3)	<0.001	75.0	(146.0)	62.9	(152.3)	52.0	(148.0)	0.645
Alcohol (g/day)	0.8	(8.7)	3.9	(20.2)	11.3	(22.2)	<0.001	0.2	(7.1)	1.7	(18.9)	9.6	(19.5)	0.001
Mediterranean diet score (rMED)	5.0	(2.0)	9.0	(2.0)	12.0	(2.0)	<0.001	5.0	(2.0)	8.0	(2.0)	12.0	(2.0)	<0.001

Numbers are counts and percentages or medians and inter-quartile ranges (IQR*). p* values obtained with Kruskal-Wallis tests for continuous variables or X^2^ tests for categorical ones. ^1^ Primary studies or less. ^2^ Waist circumference ≥102 cm (men) or ≥88 cm (women). ^3^ MET: Metabolic equivalents. ^4^ Sum of recreational and household physical activities. MD adherence levels defined as low: 0–6 points, medium: 7–10 points, and high 11–18 rMED score points.

## Data Availability

The data will be made available from the authors upon reasonable request.

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
