# Peer review of "Mediterranean Diet and Risk of Dementia and Alzheimer’s Disease in the EPIC-Spain Dementia Cohort Study"

_nutrients, 2021, doi:10.3390/nu13020700_

Round 1

Reviewer 1 Report

This manuscript explores associations between a Mediterranean-style diet (MD) and incident dementia, with a mean of 22 years follow-up. A MD was found to have a negative association with dementia risk, including after adjusting for some confounders (but possibly also for some non-confounders, see below). The analyses are thorough and, mostly, very thoughtful, but perhaps look at the data in more ways than are needed/justified (see below again) and sometimes seemed to risk over-interpretation of non-statistically significant results (see below also). In most places, the manuscript is carefully and appropriately worded, so exceptions to this (as noted below) seem like oversights to me, but the manuscript should be carefully checked for inappropriate causal language alongside any misuses of “significant”. Some changes would also be needed to satisfy the STROBE checklist.

Overall, I think that this is a very interesting and well-written manuscript and I hope that the authors and the editor find my comments useful in deciding on revisions to what I think would be a very useful addition to the literature if my concerns were addressed, which I think the authors have already demonstrated they have the knowledge and skills to do.

Some of the continuous variables described in Table 1 are clearly skewed (particularly nuts and seeds, alcohol, and coffee and tea, but also most of the others). We’d expect nuts and seeds, alcohol, and coffee and tea in particular to be positively skewed a priori and I think medians and either IQRs (as single numbers) or 25th and 75th percentiles (as pairs of values) would be more useful in describing all of the variables with meaningful amounts of positive skew. Table 1 was also silent on where the p-values came from, which should reflect the distribution of residuals from the tests/models, and I’d like to see this documented in a table note. On Line 176, you say you used MW-U and KW “as appropriate” and this further supports the non-normality within groups noted here. I’m not a fan of the expression “as appropriate” as it’s not always helpful in reproducing results (what is “appropriate” to one statistician isn’t always so for another). Here, there are two case groups and three rMED groups, so I think it would be clearer to explain which test was used when, including why parametric models were not used, particularly given you’re reporting means and SDs (Line 174), although I think the issue here will be with the descriptives rather than the tests. There are plenty of times when it’s appropriate to mix means and SDs with non-parametric tests, but this isn’t one of the “usual” means+SDs with t-tests and ANOVAs and medians+IQRs with MW-U/KW pairings, so it’s worth very briefly explaining if the current approach is retained. As a tangential comment about descriptives, In Table 1 you give the number and percentage currently smoking—was there a reason for not looking at current, former, and never smokers?

It's not entirely clear to me why both categorical and continuous versions of rMED are being investigated here, particularly since neither seems to have been identified as the primary analysis, and my apologies if I’m not picking up something that was intended to do just that. The former approach sacrifices considerable statistical power and there are many articles explaining the problems with categorising variables (including the loss of power, non-biologically-plausible [step-like] associations being modelled, and issues with replicability in new data sets with different distributions). The apparent flexibility from categorising can be obtained by adding additional terms to the continuous model, higher order polynomial or other transformations, or by using splines, as is the case here (on this note, can you explain why you’ve used knots at the two tertiles, Lines 184–185? Was there an a priori reason for this or was it empirically guided?) If there are clinically meaningful categories, such as hypertensive and overweight, a case can be made for using this for research that will be applied in a clinical setting, e.g., identifying high-risk patients for future monitoring, but I’m not picking up on such an argument here. Looking at both options seems most likely to inflate both Type 1 and 2 errors overall and makes the story less straightforward for me. Can you explain, including in the manuscript as I’m sure some readers will also wonder about this, why two different ways of measuring the exposure of interest has been used?

As is usually the case with observational research, the underlying causal model that informed the variables included in the models is absolutely critical. I’m afraid that I didn’t have a clear idea of the authors’ model, and I’m sure that they do have one, by the end of the methods and feel that this warrants some additions to the methods, and, I’d suggest, a directed acyclic graph as a supplement to clearly communicate this model. Note that Line 36’s “multivariate”, which I think should be “multivariable” (see reference [30] which uses “multivariable” rather than “multivariate”), identifies that some (presumably) confounding and/or competing exposure variables have been included in the analyses, but readers of the abstract will need to know a little more than this (and perhaps a little more than the reference on Lines 43–44 to “major cardiovascular risk factors” if space permits in the abstract) to interpret the results that follow in the abstract. At the least, I think that this description should be moved to before the results are presented in the abstract.

Given that the “univariable” analyses at the bottom of Table 1 indicate no evidence for an association between MD and the incidence of dementia (p=0.539 for the any dementia question and p=0.617 for the AD question), and given that all other analyses are adjusted, these results lead me to ask questions about the covariates included where evidence of associations does emerge (albeit for different questions given the time to event models). Some of these variables seem to me to be likely to lie, in whole or in part, on the causal pathway between MD and dementia/AD. Sex, education, and age are all clear in their roles as potential confounders and competing exposures, for example. However, energy intake (in Model 1, Table 2; and all of the models in Table 3), hypertension, hyperlipidaemia (the authors note on Page 16 that both these could be part of mechanistic explanations for MD’s effects on dementia), coffee/tea intake, and dietary variables (all in Model 2, Table 2; and all in all models in Table 3), as other examples and not an exhaustive list, all seem likely to be intervening variables to me (certainly much more than they would be confounders). It would be worth thinking whether any of these could be colliders also (e.g., energy intake decreasing from MD due to dietary choices and from loss of appetite with dementia seems at first glance to be plausible to me as one of the mechanisms that might distort the association between MD and dementia as a collider as well as as a mediator). I think that each of these variables needs some careful justification, and a DAG (with references in the text that describe and justify the DAG) would be my suggestion here. Note that you say you used STROBE (Lines 170–171) and this is covered by item 16a of the STROBE checklist, which reads: “Give unadjusted estimates and, if applicable, confounder-adjusted estimates and their precision (e.g., 95% confidence intervals). Make clear which confounders were adjusted for and why they were included.” As indicted in the final sentence, the reason for including confounders is necessary as well as listing these. A DAG would also help in identifying and justifying suggestions for further research in the Discussion.

Related to all of this, I’d like to see a Model 0 (unadjusted) added to Tables 2 and 3, as indicated in item 16a, and I suggest energy be removed from Model 1 (Table 2) if you agree that it could lie on the causal pathway. I appreciate that you report a large number of results already, but perhaps Table 3 could be split into two using the same Model 0, Model 1, Model 2 approach as Table 2?

I appreciated the examination of potential effect modifiers. I would like to see similar, and brief, justification for these variables. What mechanisms could plausibly motivate looking to see if there was evidence that a one (or two) unit increase in the rMED score might have a different association with dementia for men and women, those with more or less education, smokers and non-smokers, and those obese or not? You don’t need to be too specific in the text, but there must have been a reason for thinking that such a difference in association was possible.

I can’t help but be a little concerned about what looks to be either overfitting, or overadjusting, given the shape of the curve in Figure 2. This suggests to me, taking a causal interpretation for ease of phrasing, that having a high rMED score was highly protective right away (reducing the risk in the first year by ~90%), becoming less protective and eventually harmful (point estimates seems to be > 1 between ~7 and ~17 years), before becoming protective again. Needless to say, the small number of cases in the early stages of the trial don’t allow any robust interpretation of this association, but the fact that the underlying model used here is allowing this shape suggests to me that the model is too complex for the phenomena being studied given the data that is available. It seems to me that monotonic functions would make the most sense here, and maybe even some slightly non-monotonic associations, perhaps caused by different mechanisms that occur at different stages of life, but these wouldn’t match the modelled curve in Figure 2. By way of analogy, we can explain a J-shaped curve with respect to BMI for cardiovascular outcomes, but we would be rightly suspicious of a w-shaped curve, for example. It’s that sort of suspicion I have when looking at Figure 2. Again, I appreciate the uncertainty in the model, but it’s the shape that seems highly implausible to me, not the statistical interpretation. I could be missing an interesting possibility here, but this figure suggests to me that the model has too much freedom and might be overfitting to an outlier or set of these. While outliers seem much less likely at the later end of the study, and would have less effect there, it is perhaps possible for a few of these to induce some additional curvature in the model. Alternatively, the causal model isn’t quite right. I’ll note that the plots for men and women in Figure 1, wriggle a little the wrong way to me at times as well, but this is the price for fitting cubic functions when there is a plateau, whereas here, in Figure 2, the wriggling in an unexpected way seems particularly pronounced rather than being an imperfect approximation.

While I appreciate the rigorous examination of alternative statistical models, I’m not convinced that the model looking at competing risks should be included, even in a supplement. The need for competing risks would be with prognostic modelling, where, for example, death of another cause prior to the onset of dementia would be a “successful” avoidance of dementia. The focus, here, however, seems to be around the aetiological question of whether MD is causatively protective against dementia. If my reading here is correct, the competing risks analyses are not relevant. This reading is strengthened by Lines 149–151 where those with significant baseline comorbidity were excluded. This moves the research question for me in the direction of aetiology (in the same way such exclusions in a clinical trial might move the research question away from effectiveness and towards efficacy). It’s important to keep in mind that the models not accounting for competing risks and the models accounting for these are not sensitivity (as the different measures of MD on Lines 128–131 can be seen as) or robustness analyses of each other, the questions they ask are of a very different nature—is MD causally linked to dementia versus will MD protect me from dementia. If the question was the prognostic one, that would make the non-competing risks models wrong in a very fundamental sense. The only times I’ve ever reported both competing risk and non-competing risks results is when the literature has included what I regarded as inappropriate applications of the latter and so these were presented separately in the discussion to enable comparisons with the (in my view, incorrect) literature. I’ll stress again, the two approaches ask very different questions and I think the manuscript, and its readers, would benefit from the authors making their question of interest explicit early in the manuscript, and early in the abstract.

EPIC-Spain is a multicentre study (as noted on Line 79) and so the clustering within the three study centres warrants consideration in analyses. In places you have stratified by centre (e.g. Line 178, Table 2, etc.) and in others you seem to have adjusted for centre (e.g. Figure 2) but neither of these accommodates any clustering within centres (the degrees of freedom issue). I appreciate that with three centres, there is little that could be done (to my knowledge) about this, but it is a methods point that could be raised in the statistical methods, perhaps around Line 178.

As a final statistical comment, how did the sample size (specifically the number of events) inform the design of the study, and the decision to proceed with these analyses? I appreciate that the sample size was a given here, but in that case, there is still a go/don’t go decision to be made. If no formal calculations were performed as part of this, this fact should still be noted and I’d suggest then drawing the reader’s attention to the 95% CIs as indications of the realised power of the analyses. I wondered about the numbers of events, though, particularly for subgroup analyses. With only 52 cases in Table 4 for those more educated and what I think is well over a dozen parameters estimated (plus stratification variables), this goes well beyond the older 10 events per variable (EPV) rule from Peduzzi, et al., and even beyond the 5 EPV rules sometimes advocated for more recently. On the other hand, the most recent positions are closer to an admission that we really don’t know, in general, what EPV are needed for logistic or survival analyses. Could the authors explain, including in the manuscript, how this affected their decisions and/or interpretation of results?

I’ll also make some specific, sometimes very minor, comments below:

Line 30: I think “prevent” is an overly strong claim here, MD is associated with reduced rates rather than preventing occurrence. I think “…to reduce the occurrence of…” would be fine. There are other uses of prevent and its inflected forms that should be checked also.

Line 33: I suggest “SD” rather than “sd” here and elsewhere given it is an abbreviation. I also suggest using the same number of decimal places for both the mean and standard deviation (See also Line 210).

Line 35: Spurious hyphen in “vali-dated”.

Line 36: Note that multivariate refers to multiple dependent variables, i.e. repeated measures data. It seems that this might be “multivariable” instead. See also Lines 186, 192, and 202, the legend for Table 2, and possibly elsewhere. https://dx.doi.org/10.2105%2FAJPH.2012.300897 is a useful discussion, and promoted further discussion, of this issue.

Line 38: It would be helpful here to operationalise “high” and “low” adherence so the reader can interpret what the “20% lower risk” reflects.

Line 39: As a reader, I would assume that this was probably for linear trend (and from orthogonal polynomials), and its source should be made explicit in the abstract.

Lines 39–40: The HR and its 95% CI should be presented here for both results. It’s good that your later results already take this approach (Line 41).

Line 40: It seems possible that a reader could, at least momentarily, misread “inverse” as “opposite to previous result”. I wonder if here, and on Page 8 and elsewhere, alternative wording could be used to make this less likely. “favourable” is a word that I would have perhaps considered and “negative” might also be worth considering. Adding “also” might help as well/instead.

Line 41: I’m not sure that the notation “HR2-points” is entirely intuitive, perhaps “HR per 2-points” would be clearer for some readers?

Line 41: While I like the focus on 95% CIs, and I think the inferential results should all follow this approach, the result for men is not statistically significant and given Lines 206–207 stating that [two-sided] p<0.05 was considered statistically significant, this result should be indicated as not statistically significant. I suggest something like: “…and, while not statistically significant, in men for AD (HR per 2-points = 0.88, 0.76-1.01).” so that the non-statistical significance is communicated before the reader sees the effect size and they won’t miss the upper CI limit being 1.01.

Line 42: To claim that the association “was stronger” in one subgroup would require a test for effect modification, and the p-value here appears to be that associated with the rMED score rather than its non-statistically significant interaction with education (p=0.055 in Table 4; on first reading, I assumed the p-value reported in the abstract was itself from a test for effect modification). Given the useful HRs and 95% CIs just above, the p-value here is relatively uninformative.

Line 44: While the strength of the association numerically differs in terms of HRs, I think I’d read this, again, as claiming evidence of effect modification. None of the tests for interaction were, as far as I can tell, statistically significant, so this should, I think, be rewritten to clearly note numerical differences (which I don’t think are particularly interesting), clinically important numerical differences (which would be), or a lack of statistically significant evidence for effect modification.

Line 63: There might be a word missing/incorrect here, perhaps “…also led RESEARCHERS to explore its…”?

Lines 72–78: This is a rather long and complicated sentence. Readers might appreciate it being broken down into two (or more) sentence.

Lines 73–76: I think you could argue either way on this point about the “first step”. Evidence of protective effects from the MD in Mediterranean countries might capture a more pure form of the MD (which would be important for efficacy), but it would also increase the risk of country-specific confounding (including those factors on Line 77); looking at MD in multiple non-Mediterranean countries might change the form of MD (which would dilute the “treatment” of MD, but be important for effectiveness) but concerns about country-specific confounding would be lessened (which is related to but also distinct to external validity). I’m not disagreeing with what you’ve said here, but I think the question might be more complicated than some readers will appreciate from this part of the text. If the text remains as it is, could you give an example or two of the nuances (Line 76)?

Line 93: I wonder if this could be made clearer since the first two (donors and civil servants) are contained within the general population.

Lines 120–121: Sorry if I’m missing something, but I don’t see the difference between “grams per 1,000 kcal/day” and just “grams per 1,000 kcal” here.

Line 122: Technically “tertiles” are the cut-points (https://en.wikipedia.org/wiki/Quantile) rather than the groups defined by these cut-points. Personally, I’d say something like: “…for the first, second, and third group defined by tertiles, respectively.” here.

Line 133: I’m not sure why “Incident” is capitalised here.

Line 145: “guarantee” is a very strong claim, one that I don’t think can be justified here. In a sample of 25 thousand, are you really sure there couldn’t be even a single exception? I’m not concerned about your methods, I think the risk of prevalent dementia at baseline is low enough, it’s the use of the absolute term “guarantee” that concerns me.

Line 149: Similarly, here I’d be inclined to say “…without evidence of prevalent dementia…”

Line 153: “SD”, as mentioned above?

Line 156: To save some readers reaching for their calculator, you could give the mean follow-up time (Lines 33 and 210) here after the total time.

Line 159: I don’t think you need the comma in “history, during”.

Line 170: I’m pleased to see this and wondered if the STROBE checklist with line numbers was available as a supplement?

Lines 180–181: The definitions of “low”, “medium” and “high” should be included around here. I suggest repeating these under every table that makes use of these categories (e.g. Tables 2, 3, and 5).

Line 206: “Stata” not “STATA” (see https://www.stata.com/support/faqs/resources/citing-software-documentation-faqs/).

Line 206: I suggest making it clear that the “P<0.05” is two-sided where this is an option (e.g., not for Chi-squared tests).

Lines 215–217: Several of these variables (e.g. vegetables, fruits, olive oil, etc.) are part of the rMED score so I’m not sure that there is much value in noting that a component of a score is associated with the total score (unless compensatory effects were plausible). Similar comments apply to Table 5 and its associated text on Page 12.

Line numbers weren’t included from Page 8 onwards so I’ll try to describe the surrounding text well enough that a text search will find the line(s) I’m referring to.

Page 8, 4th line down: It’s not immediately clear to me whether this p-value is from a linear trend on the categorical version of rMED or for the linear slope using the continuous version.

Page 9: I strongly recommend against “borderline” as a qualifier for statistical significance. This particular phrasing has been discussed in the statistical literature and found to be unhelpful.

Page 10: You say “significant heterogeneity” but then give an interaction p-value of 0.055, which is inconsistent with Line 206. I also recommend against unqualified “significant”/”significance”/”significantly” as this could refer to practical/clinical as well as statistical significance.

Page 10: Similarly, “more significantly” doesn’t make sense to me (even if the p-value was lower).

Page 15: When you say “20% lower risk” (and also on Page 17), it’s not clear here exactly what this comparison is. See Lines 38–39 where it is clearer (at least in terms of the levels of the variable), and also the legend to Table 2 and the text on Page 10 about Table 4 where the comparison is made more obvious.

Page 15: When you say “There is suggesting evidence”, do you mean “suggestive”?

Page 15: Perhaps readers might benefit from being told the range of the MeDi score when trying to interpret these HRs? The thirds (see point about “tertiles”) for MeDi and MIND are probably clear enough as is.

Page 17: This is perhaps a little opinionated, but I don’t consider “large sample size” as a strength in and of itself (what is “large” is highly context dependent and also affected by measurement reliability, covariate availability, repeated measures, etc.), although if this produced sufficiently precise estimates, as, for example, shown by sufficiently narrow CI widths to enable interpretation of practical/clinical significance, that would definitely be a strength for me.

Page 17: I also felt the last sentence of the strengths was perhaps a little unclear. Could this be made more precise here as to what the exact strength you see is?

Figure 1: The meaning of the dashed lines, presumably the upper and lower 95% CI limits, didn’t appear to have been made clear in the legend, as far as I could tell.

Table 2: In the legend, I think the non-statistical significance of the categorical associations should be made clear before reporting the results given that the threshold for statistical significance was declared in the methods, rather than saying “…was associated with…” which will be interpreted as being statistical significant by some, if not most, readers.

Table 2: The order of results didn’t seem obvious to me. The categorical results are reported, followed by the continuous, with p-values for linear and then non-linear trend. If the last two are from orthogonal polynomials, which would be my expectation from simply looking at the table, they should be adjacent to the point estimates for that. If they are for the continuous version, that could be made clearer. The same applies to Table 3 and perhaps elsewhere.

Reviewer 2 Report

Comments to the authors

This is an interesting large prospective study, with long-term follow-up.

The introduction and the methodology are effective and clear. The results are presented in detail, with sufficient number of tables and figures. The discussion is adequate and includes a sufficient number of other studies on the same topic. There are also the appropriate  references, most of which are recent.

Round 2

Reviewer 1 Report

Thank you very much for your thoughtful responses to my queries and comments. As is almost always the case with large, complex studies, there are multiple options and opportunities for what I think could be regarded as valid analyses and so, in that spirit, with this review of the revised manuscript, I’ll try to avoid any dogmatism on my part and focus on a few remaining/new queries (listed below), that are all about minor wording points, around references, relating to editing from the previous version, and similar. There are a few new, but still minor, comments on unedited text and I apologise for not noting those previously. I have no substantial queries remaining and congratulate the authors for this revised version of their manuscript. It’s always a privilege to review work of this calibre.

Purely as an aside, one option for avoiding the analyses of the categorical version of rMED could have been to define “low” as prototypically 3 (mid-point of 0­–6) and “high” as prototypically 14.5 (mid-point of 11–18), giving a difference of 11.5, which would be equivalent to a HR of 0.92^(11.5/2) = 0.62. Other options could be used, including an absolutely minimum difference of 5 (11 versus 6), i.e. a HR of 0.92^(5/2) = 0.81, or using the mean or median value in each group to identify the difference of interest. Such approaches have an advantage, to my way of thinking, of making the comparison absolutely explicit, whereas “low” versus “high” otherwise depends on the distributions of scores within each group (this is similar to the problem that distributions within BMI categories can differ, “normal” versus “obese” isn’t the exact same question in all countries where the mean/median BMIs differ between countries for those groups).

Line 41: I think I understand now that “(P for linear trend = 0.021)” is for the continuous exposure version of rMED, so suggest that this is moved to be alongside the HR per 2 points instead of after the categorical comparison (where, based on the 95% CI, p>0.05), i.e. “HR of dementia was 89% (HR = 0.92, 0.85-0.99, p=0.021)…”

Line 79: I think “actually” is tautological after “really” in “really could actually” and suggest deleting one of these words here.

Line 80: Perhaps add the definite article to “where [the] MD pattern” as you do elsewhere, e.g. Lines 61 and 72 just above and Lines 141 and 251 further below.

Lines 80–81: This just seems obviously likely to be true and so I’m slightly reluctant to ask for a reference, but it does seem possible to query whether (in principle and in general) the original form is truly better preserved in all cases where the form originated.

Line 128: Missing period after “cohort”.

Line 175: I noticed a lack of an ‘Oxford comma’ before the “or” in “such as cancer, cardiovascular disease or diabetes”, whereas you seem to (at least mostly) include these, e.g. Lines 164–165 just above.

Line 190: Missing “ranges” in “interquartile [ranges] for continuous ones.” Also you say “interquartile” here but “inter-quartile” in the legend to Table 1 (and I especially apologise for the pedantry here!)

Line 191: You say “Mann-Whitney (for…” here but “Mann-Whitney U tests for…” in the legend to Table 1 and might as well be consistent again (and I should apologise again!)

Line 208–209: The clause “which could partially account for confounding as related to the exposure or the outcome or both” is perhaps vulnerable to a slightly misleading reading as confounding is always for an association involving a specific exposure:outcome pairing and, under the classical definition, requires an association with both the exposure and outcome, while not being on the causal pathway between the exposure and outcome. A lack of either of these associations means that the variable cannot be a confounder for that particular exposure:outcome association, so both are necessary/neither is sufficient. As an aside, the change-in-estimate definition of confounding coincides with the classical definition for linear models, although does not necessarily do so for logistic regression (see DOI: 10.1016/0895-4356(91)90203-L). I believe that the same problem of “mavericks” applies to HRs as well as ORs, although I don’t think I’ve seen this mentioned in the literature. In any event, assuming you are using the classical definition of confounding, I suggest rewording this text to say: “which could partially account for confounding of the exposure-outcome association” instead.

Line 214: Perhaps “g/day” should be in parentheses as you use for “(in ml/day)” on the preceding line and for METs on Line 211 (and more generally for the levels of categorical variables here and the units in Table 1).

Line 252: Should there be an “and” before “an estimated…”?

Line 288: Shouldn’t this refer to Figure S5 and not S2? If so, I’m not sure I agree with the descriptions. Neither the AD or non-AD fitted curves look monotonic (all contain at least one local extrema point to my eye). I’d also suggest rewording “suggesting a linear relationship (P for non-linearity = 0.353…)” as this could be misread as claiming the null from a non-statistically significant result, rather than noting “suggesting NO EVIDENCE AGAINST a linear relationship (P for non-linearity = 0.353…)”

Line 302: I’m too cautious for “likely” here and suggest “possible”.

Line 342: I think you mean Figure S2 rather than S3 here? (c.f. Lines 444 and 496.)

Line 344: Based on your response to my query about this, aren’t the competing risks models completely removed now? They are not mentioned in the legend to Figure S3. See also Lines 445–447 where these are referred to again.

Line 353: Perhaps quote “high” as you do for “low” on the following line?

Line 366: Sorry, I suspect the details are in reference [33] which is from 1991 and which doesn’t seem to be available online, at least not for my university, but is this a “confidence interval” (i.e., a sample-size dependent measure of precision around an estimate) or should it be a “reference interval[/range]” (i.e., a distribution of values such that those more than 2 SDs [Line 167, I note that this is SDs and not SEs] away from the mean could be identified)? Mendez (reference [23], Line 371) certainly describes reference ranges based on a normal approximation.

Line 380: Sorry for this new comment on unedited text, but perhaps change “that adherence to a MD score could” to “that adherence to MD could” as adherence makes more sense to the diet rather than a score (unless you want to define some degree of adherence based on the score).

Line 381: Should this “9%” be “8%” (as edited to on Lines 41 and 252).

Line 382 Assuming this 6% is based on “0.94 (0.86, 1.03)… P for linear trend 0.196” in Table 3, the wording here seems overly strong for a clearly non-statistically significant finding. Perhaps add a parenthetical “(although not statistically significant)” or similar to this particular result, as the 8% (assuming this is what the “9%” here represents) reduction was statistically significant?

Line 394: On re-reading this unedited text, the “9 to 10%” alongside the HR of 0.91 struck me as slightly confusing. I’m assuming the former is to also refer to their analysis after excluding AD with coexisting stroke? If so, I’d either only show the 9% (removing the 10%), add the other HR of 0.90, or qualify the HR as “with coexisting stroke included”.

Line 398: Isn’t the “for each additional point of the MeDi score (HR = 0.91; 95%CI: 0.83, 0.98)” here just repeating “by each additional point of the MeDi score (HR = 0.91; 95%CI: 0.83, 0.98)” on Lines 394–395? Or am I confusing myself?

Line 399: I see you’ve adopted my suggestion around the tertile nomenclature, so perhaps “third” rather than “tertile” here?

Line 401: Similarly, “groups based on tertiles” here to be more consistent with your wording for these in your own study?

Lines 445–447: See comment about competing risks above.

Table 5: The “2” in “X2” in the legend isn’t superscript, c.f. the same in Table 1 where it is.

Supplementary Figure 1: The spacing seems (on screen at least) to be inconsistent around equality signs, e.g. only trailing space: “N= 25,016”, both spaces “n = 1652”, and only leading space “n =24,241”. The text “Navarra: N = 8084” seemed cut-off at the bottom, at least in my copy of the figure.

Supplementary Figure 2: Sorry for not asking before, but why is this for a 1 SD change in rMED rather than the 2 units as in the manuscript?

Supplementary Figure 5: Isn’t “Non-Alzheimer’s dementia” inconsistent with “Alzheimer’s disease” here and elsewhere (e.g. Line 170 and the caption for Figure 3)?

Figures: x-axis labels for rMED are slightly inconsistent, being “Mediterranean Diet Score (rMED)” for Figures 1 and S6, “rMED score” for Figure S3, “rMED” for Figure S4 (which makes sense given the space available with the panel of 4 figures), and “Mediterranean Diet Score” for Figure S5. Note also that Figures 1 and S6 repeat the label under each sub-figure, whereas S5 includes it only for the middle of the 3 sub-figures.
